# HMaViz: Human-machine analytics for visual recommendation

Submission ID: 38

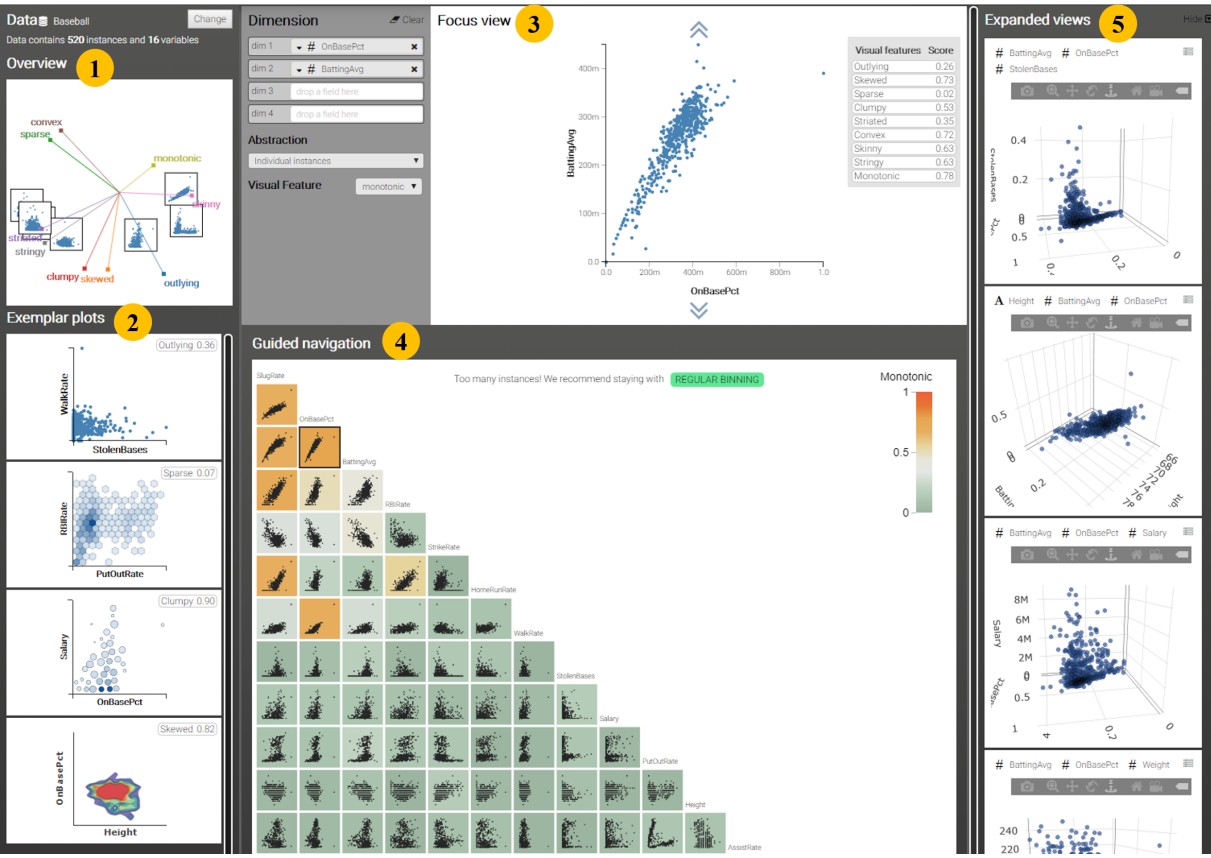

Figure 1: The visual interface of HMaViz framework: (1) Overview, (2) Examplar plots, (3) Focus view, (4) Guided navigation, and (5) Expanded view.

## ABSTRACT

Visualizations are context-specific. Understanding the context of visualizations before deciding to use them is a daunting task since users have various backgrounds, and there are thousands of available visual representations (and their variances). To this end, this paper proposes a visual analytics framework to achieve the following research goals: (1) to automatically generate a number of suitable representations for visualizing the input data and present it to users as a catalog of visualizations with different levels of abstractions and data characteristics on one/two/multi-dimensional spaces (2) to infer aspects of the user's interest based on their interactions (3) to narrow down a smaller set of visualizations that suit users analysis intention. The results of this process give our analytics system the means to better understand the user's analysis process and enable it to better provide timely recommendations.

**Index Terms:** Human-centered computing—Visualization—Visualization application domains—Visual analytics;

## 1 INTRODUCTION

Over the years, visualization has become an effective and efficient way to convey information. Its advantages have given birth to visual software, plug-in, tools, or supporting libraries [5,28,44]. Tools have their own audiences and playing fields, and they all share common characteristics; that is, no tool fits for all purposes. It is a challenging task for analysts to select the proper visualization tools to meet their needs, even for data domain knowledge experts, because of the ineffective layout design. This problem becomes more challenging for inexperienced users who are not trained with graphical design principles to choose which visualization is best suited for their given tasks.

In particular, researchers tackle this problem by providing a visualization recommendation system (VRS) [9, 31, 47] that assists analysts in choosing an appropriate presentation of data. When designing a VRS, designers often focus on some factors [49] that are suitable in specific settings. One common factor is based on data characteristics in which data attribute is taking into consideration; one example of this approach was presented by Mackinlay et al. in Show Me [33]. This embedded Tableau's commercial visual analysis system automatically suggests visual representations based on selected data attributes. The task-oriented approach was studied in [9,43], where users' goals and tasks are the primary focus. Roth

and Mattis [43] pioneered integrating users' information-seeking goals into the visualization design process. Another factor is based on users' preferences in which the recommendation system automatically generates visual encoding charts according to perceptual guidelines [38]. This paper seeks to address this problem by proposing a visualization recommendation prototype called HMaViz. Thus, Our main contributions of this paper are:

- We propose a new recommendation framework based on visual characterizations from the data distribution.

- We develop an interactive prototype, named HMaViz, that supports and captures a wide range of user interactions.

- We carry out a user study and demonstrate the usefulness of HMaViz on real-world datasets.

The rest of this paper is organized as follows. Section 2 summarizes existing studies. In Section 3 and Section 4, we describe the methodology and design architecture of HMaViz prototype in detail. Section 5 demonstrates the usefulness and feasibility of HMaViz via a case study. Challenges and future work are discussed in Section 6.

## 2 RELATED WORK

### 2.1 Exploratory visual analysis

In 2016, Mutlu et al. proposed and developed VizRec [38] to automatically create and suggest personalized visualizations based on perceptual guidelines. The goal of VizRec is that it allows users to select suggested visualizations without interrupting their analysis work-flow. Having this goal in mind, VizRec tried to predict the choice of visual encoding by investigating available information that may be an indicator to reduce the number of visual combinations. The collaborative filtering technique [21, 48] was utilized to estimate various aspects of the suggested quality charts. The idea of collaborative filtering is to gather users' preferences through either explicitly Likert rating scale 1-7 given by a user or implicitly collected from users' behavior. The limitation of this study is whether users are willing to give their responses on tag/rating for ranking visualization because these responses were collected via a crowd-sourced study, which in turn lacks control over many conditions. Another approach based on *rule-based system* was presented by Voigt et al. [50]. Based on the characteristics of given devices, data properties, and tasks, the system provides ranked visualizations for users. The key idea of this approach is to leverage annotation in semantic web data to construct the visualization component. However, this annotation requires users to annotate data input manually, which leads to the limitation of this approach. In addition, this work is lacking in supporting the empirical study. A similar approach to this study was found in [38].

As the number of dimensions grows, the browsable gallery [55, 56] and sequential navigation [15] do not scale. The problem gets worse when users want to inspect the correlation of variables in high dimensional space: the number of possible pairwise correlations grows exponentially to the number of dimensions. A good strategy is to focus on a subset of visual presentations prominent on certain visual characterizations [56] that users might interest in and a focus and context interface charts (of glyph or thumbnails) for users to select. Most recently, Draco [36] uses a formal model that represents visualizations as a set of logical facts. The visual recommendation is then formulated as a constraint-based problem to be resolved using Answer Set Programming [6]. In particular, Draco searches for the visualizations that satisfy the hard constraints and optimize the soft constraints. In this paper, our framework offers personalized recommendations via an intelligent component that learns from users via their interactions and preferences. The recommendations help users find suitable representations that fit their analysis, background knowledge, and cognitive style.

## 2.2 Personalized visual recommendations

Personalization in recommendation systems is getting popular in many application domains [27]. At the same time, it is a challenging problem due to the dynamically changing contents of the available items for recommendation and the requirement to dynamically adapt action to individual user feedback [32]. Also, a personalized recommendation system requires data about user attributes, content assets, current/past users' behaviors. Basing on these data, the agent then delivers individually the best content to the user and gathers feedback (reward) for the recommended item(s) and chosen action(s). In many cases, characterizing the specified data is a complicated process.

Traditional approaches to the personalized recommendation system can be divided into collaborative filtering, content-based filtering, and hybrid methods. Collaborative filtering [45] leverages the similarities across users based on their consumption history. This approach is appropriate when there is an overlap in the users' historical data, and the contents of the recommending items are relatively static. On the other hand, content-based filtering [35] recommends items similar to those that the user consumed in the past. Finally, the hybrid-approaches [29] combine the previous two approaches, e.g., when the collaborative filtering score is low, it leverages the content-based filtering information. The traditional approaches are limited in many real-world problems that impose constant changes in the available items for recommendation and a large number of new users (thus, there is no history data). In these cases, recent works suggest that reinforcement learning (specifically contextual bandits) are gaining favor. A visual recommendation system belongs to this type of real-world problems. Therefore, this work explores different contextual bandit algorithms and apply them in characterizing and realizing a visual recommendation system. The contextual bandit problems are popular in the literature; the algorithms aim to maximize the payoff, or in other words, minimize the *regret*. The regret is defined as the difference in the award by recommending the items (actions/arms interchangeably in terms of Contextual Bandit), which are different from the optimal ones.

## 3 METHODS

### 3.1 Data abstraction

Due to the constant increase of data and the limited cognitive load of human, data abstraction [30] is commonly adopted to reduce the cost of rendering and visual feature computation expenses [3]. Data abstraction is the process of gathering information and presented in a summary form for purposes such as statistical analysis. Figure 2 shows an example of data aggregation on two-dimensional presentations. Notice that the visual features in our framework will be computed on the aggregated data, which allows us to handle large data [13].

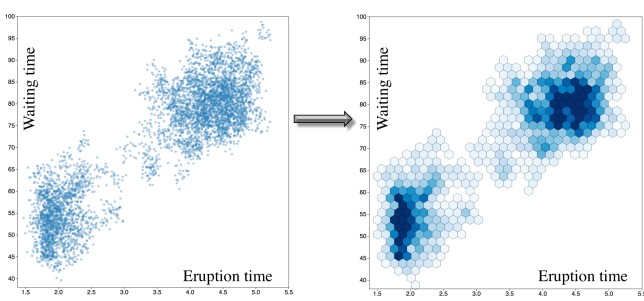

Figure 2: Abstraction of the Old Faithful Geyser data [20]: Scattertplot vs. aggregated representation in hexagon bins.

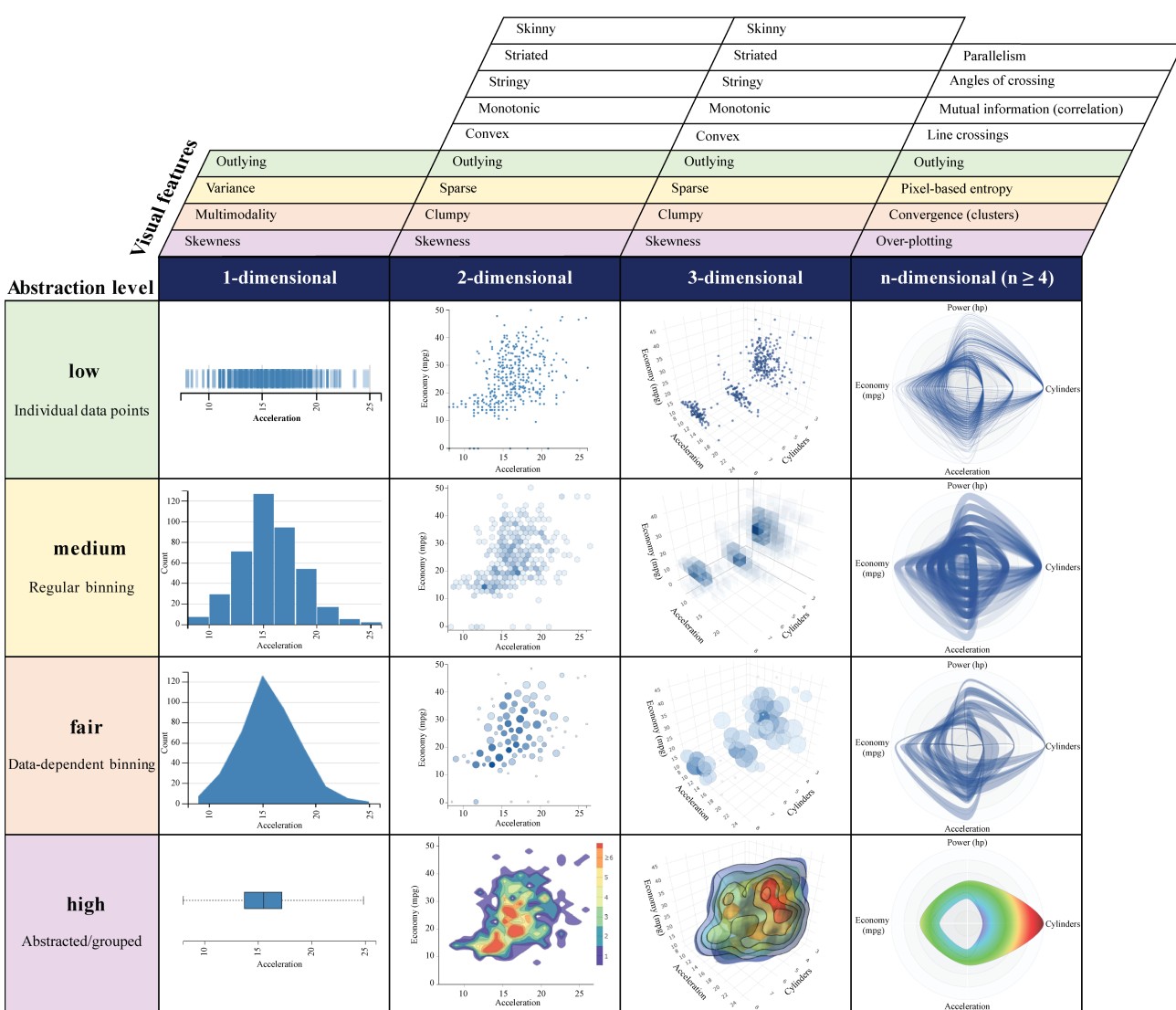

Figure 3: Our proposed HMaViz visual catalog: (top-down) More abstracted representation of the same data, (left-right) More complicated multivariate analysis, (top) The associated visual features for each type of analysis (the feature cells are colored by the abstraction level if they are plotted in the HMaViz default exemplar view).

## 3.2 Visualization catalog

In our framework, the users and visualizations are characterized on the following criteria: number of data dimensions (univariate [26], bivariate [11], and multivariate data [14, 57]), visual abstractions described in Section 3.1(individual data instances, groups [40, 41], or just summary [58]), and visual patterns (trends [23], correlations [51], and outliers [25, 52]). While each of these three dimensions has been studied extensively in the visual analytics field [3, 10], to the best of our knowledge, there is no existing framework that incorporates all together in human-machine analytics for visual recommendation system. Figure 3 summarizes the projected dimensions in our visual analytics framework: type of multivariate analysis, statistical-driven features, levels of data abstractions, and visual encoding strategies.

## 3.3 Learning algorithm

Building upon the first task, the second task focuses on the visual interface that can capture the users' interest [8]. We first explored the four mentioned algorithms for contextual bandit problems, namely

$\varepsilon$-*greedy*, UCB1 [2], LinUCB [32], and Contextual Thompson Sampling. We defined our problem following the k-armed contextual bandit definition discussed in the previous section. Also, in our case, the reward is the combination of (1) whether the user clicks on the recommended graph and (2) how long after clicking the user spends on analyzing the graph. In this case, clicking on the chart is not enough, since right after clicking, the user may use the provided menu to modify the recommended item. For instance, after clicking on a graph, the user uses the given menu to change the abstraction level of it (e.g., from individual point display to clustering display). This change means the agent does not recommend the appropriate abstraction level (though the other features might be correct).

After defining the problem, we generate a set of simulated data according to the number of variables, variable types, abstraction levels, and visual features for each graph to test the regret convergence of the four algorithms. These simulated tests help in selecting an appropriate algorithm for our solution or change of the required data collection to reflect the user's behavior [22]. As from experimental results, LinUCB and Thompson Sampling gave better results

compared to other algorithms. Notably, LinUCB outperformed the other algorithms on the simulated data. Thus, we select it to build the learning agent in the current implementation. Note that these experimented results do not necessarily imply Thompson Sampling method is not as good as LinUCB for the actual or different datasets or different sets of parameters (that we haven't been able to explore exhaustively) for the Thompson Sampling algorithm. Therefore, the learning agent itself is developed as a separate library with a defined set of interfaces detailed in Section 4. This separate implementation allows us to replace the learning agent using a new algorithm or applying the algorithm to different recommendation tasks in the system without having to change the system architecture much.

## 4 HMaViz ARCHITECTURE

Before applying machine learning techniques or fitting any models, it is important to understand what your data look like. The system generates a diverse set of visualizations for broad initial exploration for one dimension, two dimensions, and higher dimensions. Lower dimensional visualizations, such as bar charts, box plots, and scatter plots shown in Figure 3 are widely accessible. As the number of dimensions grows, the browsable gallery [55, 56] and sequential navigation [15] do not scale well. Therefore, our framework provides two unique features to deal with large, complex, and high dimensional data. First, we use statistical-driven components that characterize the data distributions such as density, variance, and skewness (for 1D), shape and texture (for 1D), convergence and line crossings (for nD). Second, we propose the use of 4 abstraction levels in our Human-Machine Analytics: individual instances, regular binning, data-dependent binning, and most abstracted (such as min, max, and median). On the human side, this helps to capture their level of interest in the data (individual, groups, or overall trend). On the machine side, the framework automatically adjusts the level of abstraction in the recommended view to render the larger number of plots (can be requested by the users) as the number of views can be exponentially increased by the number of variables in the input data [34].

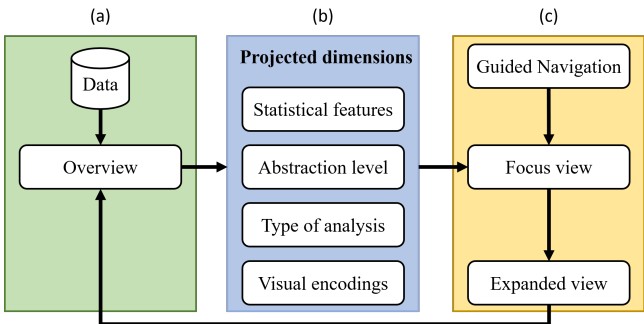

Figure 4: Flow chart of HMaViz: (a) Overview panel, (b) Recommended views projected on the four dimensions: statistical-driven features, abstraction level, type of multivariate analysis, and visual encodings (c) Guided navigation view and expanded view.

### 4.1 Components of the HMaViz

Figure 4 shows a schematic overview of HMaViz. After data is fed into the system, the statistical-driven features are calculated and plotted on the overview panel 4(a). From the overview panel, heuristically defined initial views are shown (i.e., ticks plot, bar chart, area chart, box plot for 1D). Recommended views are projected on the four dimensions, as shown in Figirue4(b) (statistical-driven features, abstraction level, type of multivariate analysis, and visual encoding) to convey users' interest. Users may select to change one or more dimensions in the interface, which may lead to the partial or full updates of the recommendation interface. For example, if

users are interested in a more abstracted representation, the guided navigation, focus view, and expanded view in Figure 4(c) need to be updated. While increasing the number of variables in the analysis might lead to the updates of overview and exemplary plots. The visual features for the next level of analysis will be calculated as well (via another web worker).

#### 4.1.1 The overview panel

Figure 5(a) summarizes the input data in the form of Biplots [19] chart, which allows users to explore both data observations and data features on the same 2D projection. From the center point of the panel (the intersection among all connected lines), the horizontal axis represents the primary principal component, while the vertical axis represents the second principal component. Each observation in the data set is represented by a small blue circle that has a relative position to the principal components. Each vector is color encoded [39].

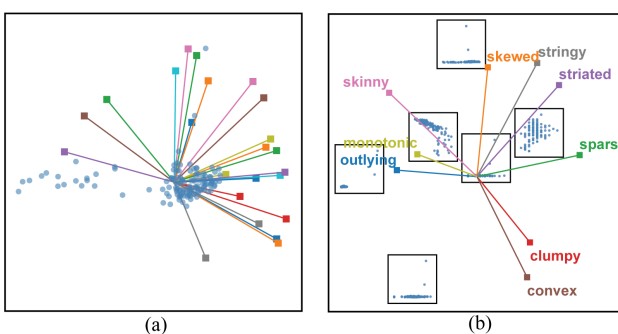

Figure 5: The overview panel of HMaViz: (a) 1D Biplot (b) 2D Biplot.

Figure 5(b) shows nine feature vectors of 2D projections, including convex, sparse, clumpy, striated, skewed, stringy, monotonic, skinny, and outlying [53]. Example plots are chosen based on their values on each of the statistical measures to covey possible data patterns in the data. The position of each thumbnail is relative to principal components. Users can start their analysis process by picking up a variable from a list, from in the overview panel, or exemplary plots explained next.

#### 4.1.2 The exemplary plots

To avoid overwhelming viewers with a large number of generated plots, we automatically select exemplary plots which are prominent on certain visual features, such as skewness, variances, outliers [12] (for univariate) and correlations [46], clusters [4], Stringy, Striated [53] (for bivariate) among other high dimensional features [14, 17]. We also heuristically associate the visual features and abstraction levels in these four exemplary plots. The predefined associations are color-coded in our catalog in Figure 3.

For univariate, HMaViz heuristically defines four levels of visual abstraction vs. four data distribution features in the initial view, including low-outlier, medium-multimodality, fair-variance, and high-skewness. The first abstract visual type (as depicted in Figure 6(a)) is the ticks plot of variable *SlugRate*, which has the highest outlier score (on the top right corner of the plot). The ticks plot is at the lowest abstraction level because every single data instance (including outliers) is plotted and selected (to see its detail). The capability is desirable in many application domains as outlier detection is one of the critical tasks for visual analysis [7].

We use the bar chart as a recommended visual abstraction for the second level (as illustrated in Figure 6(b)) because we want to highlight the skewness of data distribution. The highest skewness value is calculated from values in a given dimension. In contrast to regular binning, the data-dependent binning starts out where the

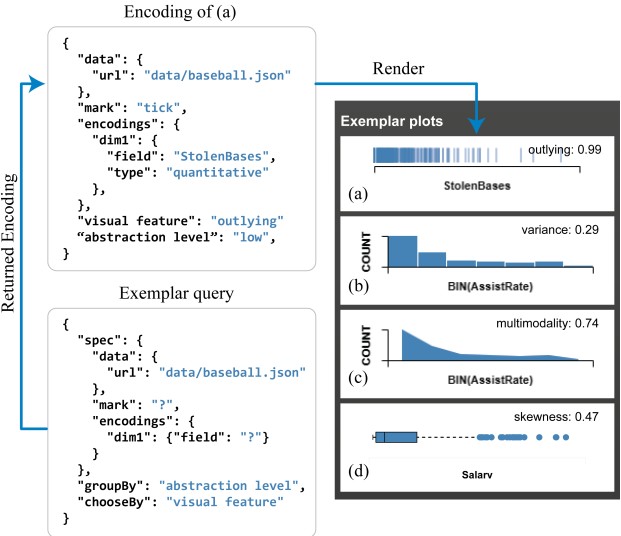

Figure 6: Univariate exemplar plots for the Baseball data: (left) Declarative language and (right) Visual representations.

actual data located and create a smooth representation of the distribution density [24]. An area chart is used for this purpose (in 1D) as the fair visual abstract type (in Figure 6(c)). The Box plot is recommended for the highest abstraction level type of visual encoding in Figure 6(d) as it is a standardized way of displaying the data distribution of each variable based on the five-number summary: minimum, first quartile, median, third quartile, and maximum [42]. We try to keep this Miller magic number consistently across the highest level abstractions (for multivariate analysis) in our framework. For example, our 2D contours (the most abstracted bivariate representation in HMaViz) are separated into five different layers.

### 4.1.3 The guided navigation

To support ordering, filtering, and navigation in high dimensional space, we provide focus and context explorations. In particular, thumbnails and glyphs [16] are used to provide high-level overviews such as Skeleton-Based Scagnostics [34] for multivariate analysis and support focus and context navigation (highlighting the subspace that the user is looking at). The guided navigation view provides a high-level overview of all variables and allows users to explore all possible combinations of variables. The view is color-coded by the selected statistical driven features and order the plots so that users can quickly focus on the more important ones [1]. Within the guided navigation panel, users can change abstraction levels as well as the visual pattern of interest.

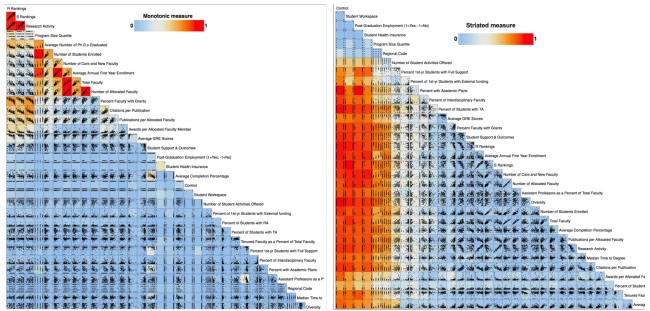

Figure 7: The navigation panel for 33 variables ordered and colored by (left) pairwise correlations and (right) Striated patterns.

Voyager [55] and Draco [36] provide interactive navigation of a gallery of generated visualizations. These systems support faceting into trellis plots, layering, and arbitrary concatenation. Our HMaViz incorporates faceted views into the expanded panel and also supports more flexible and complicated layouts such us biplots, scatterplot matrices (as depicted in Figure 7), and parallel coordinates to provide visual guidance via data characterization methods [18, 54].

### 4.1.4 The expanded view

*From one dimensional to two-dimensional visualization.* Figure 8(a) shows the recommended scatterplot when the current visualization is a tick plot (since every instance can be brushed in both plots). If the focused plot is a bar chart, then the suggested chart is the 2D hexagon bins, as depicted in Figure 8(b) (since they are both in the medium abstraction level). When the area plot is used, the recommended representation is the 2D leader plots, as depicted in Figure 8(c). The leaders (balls) are representative data points that groups other data points in their predefined radius neighborhood [24]. The intensity of the balls represents the density of their cluster, while the variance of their members defines the ball sized. And finally, we use the contour plot as the next level recommendation of the box plot, as shown in Figure 8(d), where the second dimension is selected based on the current selected visual score.

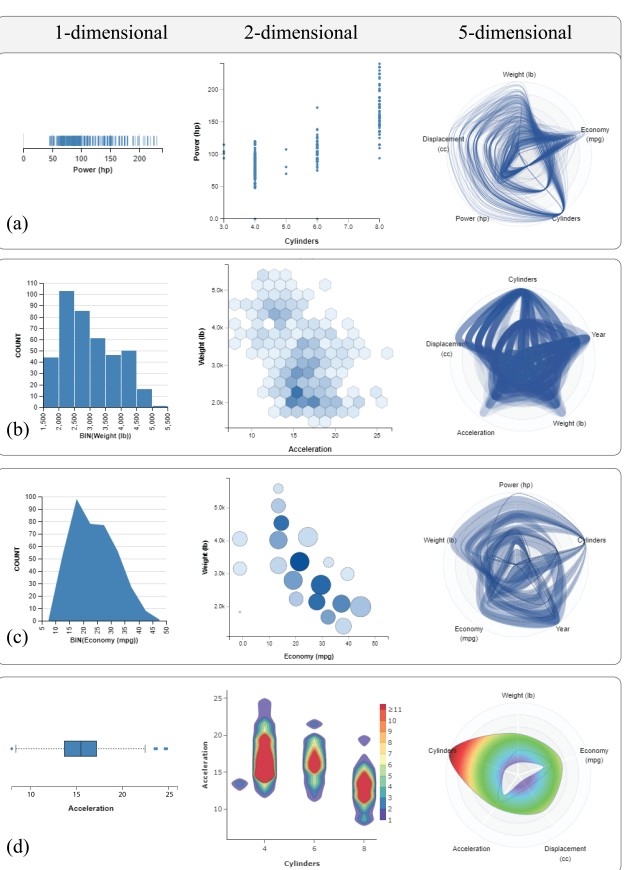

Figure 8: Visualization recommendation from 1D to 2D and from 2D to nD: Plots in the last row are the highest abstraction. Notice that variables in each plot are different.

*From two-dimensional graph to higher-dimensional graph.* The rightmost column in Figure 8 shows examples of equivalent higher-dimensional representation for the ones on the left. Notice that in the right panel of Figure 8(c), the closed bands (groups) have different widths as the variance in these groups varies on each dimension.

Figure 8(d) presents our new radar bands, which summarize the multivariate data across many dimensions. In particular, the inner and outer border of the bands are the first and third quartiles of each dimension — the middle black curve travel through the medians of the dimensions.

### 4.1.5 The learning agent

We apply reinforcement learning in our framework to learn and provide personalized recommendations to individual users via their interactions and preferences. As discussed in Section 3, we implemented our learning agent as a separate library before deploying to our target application. This separation makes it applicable in different recommendation tasks in our application and also can be easily replaced by a different algorithm without impacting the overall system architecture.

Figure 9 shows the main components of our learning agent implementation. The first task is to create a new agent. After creating, the newly created agent can have options to learn online and offline. In online learning mode, the agent first observes the user context and combine that with the data of the visualizations available for recommendation. It then uses its learned knowledge to estimate the scores for each of the available graphs and recommend to the user the items with higher estimated scores. After recommending, the agent observes the rewards from the user. In our case, the rewards mean the combination of (1) whether the user clicks on the graph and (2) the number of minutes the user spends on exploring that graph. After having the actual rewards for the recommended visualizations, the agent updates its current knowledge from this trial.

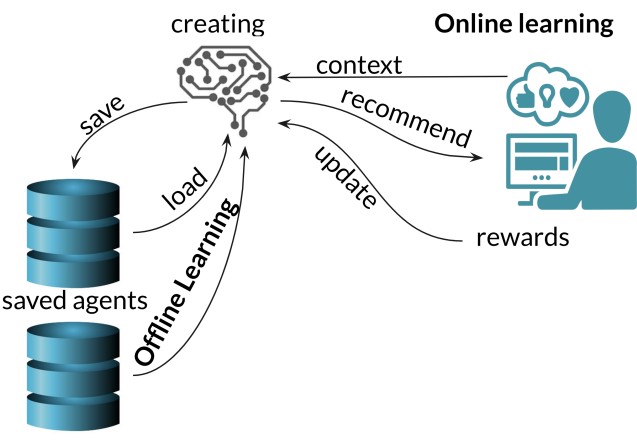

Figure 9: Components of the Contextual Bandit learning library.

On the other hand, in the offline training mode, there is a recorded set of $T$ trials; each trial $t$ contains a set of $K$ graphs with corresponding $d$ context features and also the corresponding rewards for that trial. The agent makes use of this offline dataset and runs through each trial to extract the better reward estimation for any given user context. Finally, it is crucial to be able to save and transfer and reload the learned knowledge from the agent. Therefore, we also provide options to save and reload the agent's learned knowledge. This transferable knowledge and also offline learning capabilities allow us to change the agent algorithm, and/or the agent can learn from available data coming from different sources.

### 4.2 Implementation

The HMaViz is implemented in javascript, Plotly, D3.js [5], and angularJS. The online demo, video, source codes, and more use cases can be found on our GitHub project: `https://git.io/Jv30Y`. The current learning agent (called *LinUCBJS*) is implemented in

JavaScript. Also, Firebase [37] is used to store data for the agent. Figure 10 depicts the steps involved in one trial in the online-learning of the agent in this current version. First, the agent reads the user-profiles and the features of the recommended graph. It then recommends a set of four graphs with corresponding IDs, which then be presented to the user. The system then monitors if the user clicks on the recommended graphs and how long the user stays in exploring the graphs to generate corresponding awards (in minutes). Finally, the agent uses these awards to update its knowledge.

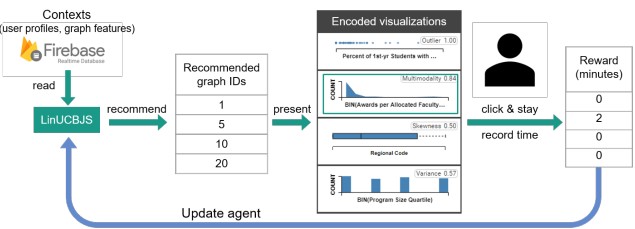

Figure 10: LinUCBJS deployment in the HMaViz implementation.

## 5 CASE STUDY

To evaluate the effectiveness of the HMaViz, we conduct a study with ten users coming from various domains. Of the ten users, three users are with statistical expertise, two users are in the psychological department, two users are in civil engineering, and three users come from the agriculture department. No member of the proposed framework is included in this study to avoid bias. The purpose of varying users is that we want our system to be viewed from multiple perspectives.

The goal of this study is to capture the user's opinions on the current system. The advantages, drawbacks of using the learning interface are addressed and iteratively refined during the experiment. The qualitative analysis method is mainly used in this study. We separate the experiment into three phrases:

- **Phase I**: Understanding the interface or how the visual framework works. Before users start exploring the dataset, they are introduced to the basic components and elements of the system, how they are connected. Navigation to traverse from one panel to another is illustrated. This phase is anticipated to take approximately 20 minutes.

- **Phase II**: Exploring data set. This phase involves users in the active experiment with 14 datasets such as Iris, Population, Cars, Jobs, Baseball, NRCMath, Soil profiles. Most of the datasets are public that can be found on the internet. We try to vary the dataset as much as possible while maintaining user familiarity with some datasets.

- **Phase III**: Gathering information. Information is gathered in both previous phases (Phase I and Phase II) for analysis and post-study. After users finished their experiment, we provide them with some opened-ended and focused questions. All data is used for analysis.

### 5.1 Findings and Discussions

**Interesting trends or patterns**. Each user has a different view of data, so similar patterns are interesting to ones but may not the case for the others. We filter out the least mentioned interesting cases and only keep the top favorites in our report. Fig. 11 illustrates some typical findings reported by most users. In Fig. 11(a), users find interesting patterns in the Baseball dataset, which is the outlying point. This point is represented by a small red circle. It is noticed that the outlier can be recognized easily from the one-dimension

chart (i.e, SlugRate), but in some situations, this outlier point is not *outlier* when taking into account in high dimensional space. However, in our case, adding one or two dimensions, the outlier stands out.

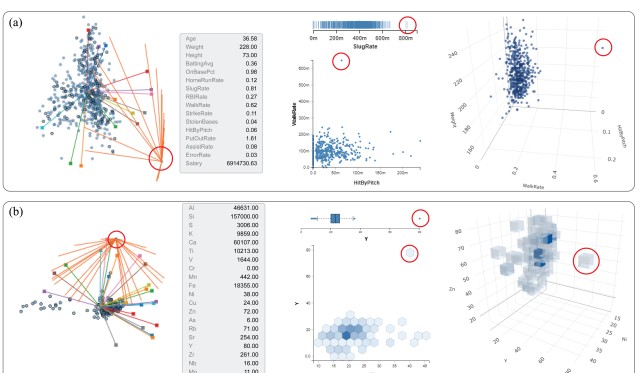

Figure 11: Exploratory analysis of detecting outliers in three navigation steps from overview, focus view, and recommended views using statistical-driven measures: a) The Baseball data b) The soil profiles.

Regarding the visual abstractions, the three agriculture users were interested in the medium level of abstraction, especially the 2D hexagon plots, since the rendering is much faster. The visual features received unequal attention from users. For example, many users will start with finding outliers in the data and looking for the best projections to highlight and compare the case.

## 5.2 Expert Feedback

For experts coming from various domains, we observed that they have various interests. Therefore, the types of analysis (and visual encoding, statistical measures, and level of abstractions) vary significantly. For examples, the three soil scientist was mostly interested in 2D projection and the correlations of variables. They specifically mentioned that they rarely go more than 2D in their type of analysis and unfamiliar with 3D charts and radar charts. They found that the guided scatterplots matrix colored and ordered by our *Monotonicity* measure is useful and visually appealing. They can easily make sense of the correlation of different chemical elements (such as Ca, Si, and Zn) and therefore differentiate and predict the soil classification. In terms of visual abstraction, they all agreed that the 2D contour map could quickly provide an overview of chemical variations. In brief, for the soil scientist, we can project their interest in our framework dimensions as:

- Type of analysis = bivariate (2D scatterplot)

- Level of abstraction = high

- Visual encoding = area (contour)

- Statistical feature = monotonicity (Pearson correlation)

Besides the positive feedback, expertise also pointed out the limitations of HMaViz. For instance, the 2D visual feature (or Scagnostics) is not a user-friendly option. Adding the animated guideline or graphical tutorial will be helpful in this case so that they can go back and forth to find out what visual features are available in the current analysis (to guide their exploration process) and the meanings and computations of each measure.

## 6 Conclusion and Future work

This paper presents the HMaViz, a visualization recommendation framework that helps analysts to explore, analyze, and discover data patterns, trends, and outliers and to facilitate guided exploration of high-dimensional data. The user indicates which of the presented plots, abstraction levels, and visual features are most interesting to the given task; the system learns the user's interest and presents additional views. The extracted knowledge will be used to suggest more effective visualization along the next step of the user interaction. In summary, we provide view recommending (with different types and complexity) and guidance in the data exploration process. Our technique is designed for scaling with large and high-dimensional data. Also, the learning agent is designed as a separate library with a clear set of interfaces. Thus it is open to incorporating new learning algorithms in the future. Especially when more user data become available, different learning algorithms should be explored and validated for personalized recommendations.

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
