# OpenReview forum: "HMaViz: Human-machine analytics for visual recommendation"
_graphicsinterface.org/Graphics_Interface/2021/Conference — Submitted to GI 2021_

### Official Review · AnonReviewer3 · 2021-01-13
**Review of HMaViz**

**Rating:** 5
**Confidence:** 5

**Review:**

This paper tackles the problem of visualization recommendation. A framework is proposed to automatically generate visualizations from input data based on a number of dimensions. The recommendation challenge is formulated as a contextual bandit problem, and different contextual bandit algorithms are evaluated. An interactive prototype is developed based on the framework, and a user study was done to evaluate the system HMaViz.

Visualization recommendation systems need to answer multiple questions: what data variables to show? What visual encodings to use? These questions, as acknowledged in the related work section, have been explored in previous work. This paper offers some new perspectives on the recommendation problem. I think the focus on data abstraction level, and the contextual bandit problem angle are interesting and novel in this paper.

Unfortunately, the current submission lacks enough technical details and explanations for me to evaluate the effectiveness of the proposed framework and the algorithms. First of all, the visualization dimensions/criteria discussed in Section 3.2 and Section 4 do not match. Section 3.2 talks about the number of data dimensions, data abstraction and visual patterns; section 4 talks about statistical distributions (is it the same with visual patterns?) and data abstraction, the number of data dimensions is left out. Secondly, it is important to explain what contextual bandit problem is. Not everyone is familiar with this term. Providing background information on how the learning algorithms work is also necessary. In addition, the paper should include information on how the simulated data was generated, and how the algorithms were evaluated. Without such critical information, it is impossible to assess the soundness of the technical contribution, which arguably is the most interesting part of the paper.

It is also unclear how the recommendation engine works when there is little training data. Given that the paper claims to use a reinforcement learning approach, the algorithm might be able to start without a lot of data, and can adapt as new data comes in. But in the very beginning when there is no user interaction, how is the recommendation made? How are different types of user actions (e.g. changing data dimensions, changing visualization parameters, exploring visualizations) used to feed into the learning algorithm? These details are important for readers to understand how well the recommendation engine works.

The case study is not very informative to me. I couldn’t tell how well the system works, and if any potentially useful visualizations are omitted by the recommender.  A comparative study with systems like Voyager and Draco would be insightful.

---

### Official Review · AnonReviewer1 · 2021-01-13
**My review**

**Rating:** 4
**Confidence:** 4

**Review:**

This paper presents an interactive prototype HMaViz which recommends to the user visualizations based on the user’s data that they are trying to visualize. The system is based on a new framework proposed in the paper, and the recommendations are made using reinforcement learning, more specifically contextual bandit algorithms.

As noted on page 3 of the paper, the framework characterizes “users and visualizations” on a number of criteria (number of data dimensions, visual abstractions, and visual patterns). The authors mention that although each of these criteria have been study separately, no existing framework has incorporated them together for visual recommendations. [Aside: I am not clear how these criteria characterize users.]

Finally for contributions, the paper claims a user study that “demonstrates the usefulness of HMaViz on real-world datasets”.

As someone who is familiar with viz research, but is in an adjacent field, I will admit that I struggled with the contributions being claimed. It is clear that the authors have done a tremendous amount of work that culminates in this paper, but the key contributions and clarity on how others will benefit from this work were not sufficiently evident to me.

If other researchers have looked at all these same criteria, but not put them together in a framework, perhaps there is a reason. Just because there is a gap in the literature, it doesn’t necessarily mean that filling the gap will be a contribution. I would have liked to see more coverage on why this is a problem that needs addressing (Why do we need a framework?). I believe my desire for more coverage here stems at least in part from the Introduction, which I found to be overly brief. The Introduction mentions in the first paragraph that it is challenging for analysts to select the proper visualization tools to meet their needs because of “the ineffective layout design”. From there on this is referred to simply as “the problem” with no further explanation. What is the ineffective layout design? How big of a problem is it?

The intro goes on to indicate that others have tried to tackle this problem, but I am left unclear on why HMaViz, and the framework it is based on, are expected to work better than prior work. The related work section doesn’t help me contextualize the contribution much better, unfortunately. Some of this comes in a short paragraph at the top of the righthand col on page 5 – a comparison with the Voyager and Draco systems, but it is short and I feel that it comes too late.

It is possible that some of my confusion may stem from the clarity in the writing. I am very sympathetic to non-native English writers, and in many places in the paper I can infer what I believe the authors intended. But there are a good number of places where it is unclear to me. For example, on page 2, in reference to a rule-based system system/approach it says “In addition, this work is lacking in supporting the empirical study.” Does this mean that there is no user study that supports/validates the approach? It goes on to directly say “A similar approach to this study was found in [38].” But I am confused, because I thought it said there was no study. Another example, also on page 2 is referring to Figure 2 and says “Notice that the visual features in our framework will be computed on the aggregated data, which allows us to handle large data.” Am I supposed be noticing/seeing this in Figure 2? If so how? Or did the authors mean to say “Note that…” which has a very different meaning.

I applaud the authors for conducting a case study with 10 users, and that they went to clear efforts to have a varied sample of participants. However, I was left quite unclear on what was actually done and how the data was analyzed. The study is referred to as an experiment but it is also noted that “the qualitative analysis method is mainly used”. Which qualitative analysis method? What were the conditions in the experiment? What did the users actually do? The Phase II description does not help me here. A key claim of the framework/system, as I have understood it, is that it learns from users’ interactions. How was this validated in the user study?

I do see that the authors have done a lot of other work, and so although they claim the user study as a contribution of their work, I don’t believe it is the main contribution, nor should the bar be terribly high. It is one piece of several in this research. But even with that said, I believe the write-up of the study as it is now is below bar – both the description of the study and the analysis.

Other/minor:
The resolution of the figures seems too low. While I can see most things okay, there are some elements that I want to be able to read (axes on the graphs for example), and even when I zoom in, the figures are just fuzzy.

There is mention that “the next level of analysis will be calculated as well (via another web worker)”. Perhaps I missed it, but where do web workers come in?


Overall, to summarize, there seems to be a substantive amount of work that went into this research. It is somewhat unclear to me, but it is possible that a substantial iteration on the writing in the paper (framing, contextualizing contribution, clarity) is all that is needed to make the contributions come through more clearly. However, it is possible that more actual work is needed, even before the iteration on the writing.

I hope these comments are helpful and wish the authors success with their work.

---

### Official Review · AnonReviewer2 · 2021-01-14
**Many things unclear**

**Rating:** 4
**Confidence:** 3

**Review:**

This paper describes a web-based infovis system that suggests visualizations by recasting the recommendation problem into something called a “k-armed contextual bandit problem” and using an existing algorithm. I was able to try the online demo linked from the paper, and it seemed to work (though I wasn’t sure how to use it) and clearly is the result of a significant amount of development effort.

The main issue is that it isn’t entirely clear what the contribution is (is it the recommendation algorithm? the web application/system? a way to categorizing viz? new types of viz like the overview panel? understanding how people select and use different viz?). Related to this, it’s not clear how this work improves on previous approaches, systems, etc., especially previous viz recommendation systems like those listed in paragraph 2 of the intro. Finally, it’s not clear that the method(s)/system/techniques were really tested and validated in the study.

These issues might be a case of writing and presentation, perhaps there’ some great work that can’t be understood in the current paper form. I’m not an infovis expert, so I’m sure I’m missing infovis research context that might have helped me fill in the blanks regarding some of the issues above. But GI is a general conference, so GI papers need to be clear enough that people who are out-of-area can still understand the basic things: what was done, how the everything works, why it’s novel, and that it was validated. The issues above suggest this isn’t the case. A careful full rewrite of the paper, likely with a revised and expanded study, might be able to address the issues. Unfortunately, as submitted, I would not argue for accept.

- - -

The second paragraph of the introduction names and cites past examples of what appear to be similar visualization recommendations systems, including some that also seem to recommend based on user goals and tasks and user preferences. But the descriptions are very brief, really only placing them in high level categories. Only [38] is included in the related work section with a few sentences of description, but even here, there is no clear comparison or explanation what is novel.  The rest of related work seems focused on recommendation algorithms or systems not mentioned in the intro as most similar systems.  This makes it very hard to understand exactly how this relates to, and is an improvement over, previous work.

Since the majority of related work is about underlying recommendation algorithms, I expected to see this as a major contribution with a clear description of it. Section 3.3 and 4.1.5 have some description about this, but it is quite vague. In section 3.3., I understand that the recommendation seems to be based on what graphs the user clicks on and how long they study them (stated in paragraph 1), but how this works in terms of the algorithm isn’t explained. The second paragraph describes how simulated data sets were used select an algorithm, but this isn’t clear either.  In section 4.1.5, the first part repeats some of 3.3 with an obvious ML toolkit level explanation in paragraph 2. I wasn’t able to understand what paragraph 3 is saying beyond that the model is trained, it runs each time the user does something (I think that’s what a “trial” means here), and that the model can be saved.  Figure 9 is so high level and generic that it could describe any ML system.

I think section 3.1 and 3.2 are explaining a framework for categorizing different kinds of visualizations. The categorization in figure 3 makes good sense from as high level, but I’m not sure what is new. The user study in section 5 seems to return to this focus on visualization types rather than the system or recommendation algorithm.

The system itself needs to be explained with enough detail that the reader can understand how it works and why is has new and better ideas. I read section 4.1 twice, and I still can’t understand how the interface works (e.g. how a user navigates the system). For example, the overview plot seems to be central to the interaction flow, but I was never able to figure out what it is visualizing or how the user interacts with it. For the most part, this section seems to focus on describing different variations of visualizations (exemplary plots) which again relates back to Figure 3 and section 3.1 and 3.2.

The study results seem to focus on user interpretation of data from different kinds of graphs (e.g. noticing outliers or not, depending on number of dimensions). I think finding outliers was a specific study task, but this isn’t stated. The second paragraph and first part of 5.2 are about types of plots preferred by participants from a specific domain. In either case, it isn’t clear to me how this validates the system or recommendation algorithm. Perhaps this says something about what I’m calling the “framework” of viz types (figure 3), but I’m not sure.  The last part of 5.2 is the only direct result about the web application itself, but it isn’t too positive. Given the lack of clarity in describing the system earlier, I’m not sure this a main part or even something that is new and considered a contribution.

---

### Meta-Review · Area_Chair1 · 2021-01-15

**Recommendation:** Reject
**Confidence:** 5

**Metareview:**

The three reviews for this paper are very consistent in both their recommendations and their comments. They all recommend that this paper not be accepted in its current form. The main issue is that there are many places in the paper that are insufficiently clear on what was actually done, rendering the contributions of the work to be insufficiently clear.

---

### Decision · Program_Chairs · 2021-01-16

Reject